# Efforts to Minimise the Bacterial Genome as a Free-Living Growing System

**DOI:** 10.3390/biology12091170

**Published:** 2023-08-25

**Authors:** Honoka Aida, Bei-Wen Ying

**Affiliations:** School of Life and Environmental Sciences, University of Tsukuba, Tsukuba 305-8572, Ibaraki, Japan

**Keywords:** genome reduction, growth fitness, minimal genome, experimental evolution, culture medium, machine learning

## Abstract

**Simple Summary:**

Genome reduction is a top-down approach to achieve the minimal genetic information essential for a living cell. The most popular and well-studied cell model for genome reduction was performed with *Escherichia coli*, advantageous for its rapid growth and easy manipulation. This mini-review discusses genome-reduced *Escherichia coli* by paying particular attention to growth fitness, the fundamental property of life. It recapitulates the efforts to minimise *Escherichia coli* genomes, focusing on experimental and computational achievements in the systematic understanding of being alive.

**Abstract:**

Exploring the minimal genetic requirements for cells to maintain free living is an exciting topic in biology. Multiple approaches are employed to address the question of the minimal genome. In addition to constructing the synthetic genome in the test tube, reducing the size of the wild-type genome is a practical approach for obtaining the essential genomic sequence for living cells. The well-studied *Escherichia coli* has been used as a model organism for genome reduction owing to its fast growth and easy manipulation. Extensive studies have reported how to reduce the bacterial genome and the collections of genomic disturbed strains acquired, which were sufficiently reviewed previously. However, the common issue of growth decrease caused by genetic disturbance remains largely unaddressed. This mini-review discusses the considerable efforts made to improve growth fitness, which was decreased due to genome reduction. The proposal and perspective are clarified for further accumulated genetic deletion to minimise the *Escherichia coli* genome in terms of genome reduction, experimental evolution, medium optimization, and machine learning.

## 1. Introduction

The genome is the blueprint of life, including all the living organism’s information. The abundance of genetic information a genome contains is nearly determined by the number of base pairs encoded on the genome, both the genes and the non-coding regions. The quantitative evaluation of the genetic abundance is known as the genome size, which largely varies among organisms. In general, the more complex the structure and ecology of an organism, the larger its genome size. For example, a typical virus has a tiny genome size of about 1~30 kbp [1], while a human cell has a genome of 6.4 Gb as diploid, and one of the ferns, *Paris japonica*, has a giant genome of 597 Gb as octaploid [2,3]. It is known that the genome size range in archaea and bacteria is between 100 kbp and 16 Mb, which can be linked to the microorganism’s ecosystem type (aquatic, host-associated, or terrestrial) [4]. The genome size roughly represents the abundance of the genetic information required for its holder cell.

Living cells are maintained by transcribing genes encoded in the genome into mRNA, followed by their translation into proteins, which shape cells and catalyse various chemical reactions. The relationship between the genome and the phenotype, the growth fitness, has often been investigated. In addition, genomes have reached their current form through mutations and horizontal gene transfer during evolution; thus, it is crucial to examine genomes to unravel the evolution of organisms. The genome size is the consequence of evolution, which either increases or decreases the genomic sequences. What the minimal genetic information needed is for life on Earth remains an open question.

## 2. Genetic Requirement for Minimal Genome

### 2.1. Genome Reduction

*Escherichia coli* (*E. coli*) has been widely used in synthetic biology due to its fast growth in rich media [5] and high transformation and plasmid integration efficiency [6]. Nevertheless, the essential and substantial genetic requirements of *E. coli* are not fully understood. Although *E. coli* has been well studied in genetics and bioengineering, and its first genome sequence was determined approximately 25 years ago [7], the molecular and physiological functions across the genome have not been fully clarified. Most *E. coli* genomes range from 3.8 Mb to more than 6 Mb, and the average is around 5 Mb. *E. coli* K-12 MG1655, one of the most well-known *E. coli* strains, has about 4.6 Mb and contains approximately 4400 genes, highly interacting with each other and shaping various biological networks. Gene regulatory network (GRN) and transcriptional regulatory network (TRN) were constructed to know the regulatory relationship of genes and often reconstructed to improve the prediction of gene expression [8,9,10,11]. Connecting GRN to the metabolic network, which is consisted of metabolites, was implemented to understand the cell system [12]. It is also indicated current GRN may not be able to predict gene expression [13]. A number of genes still have unknown functions, and understanding the entire cell as a dynamic system is challenging due to the complexity of genetic and metabolic networks. In other words, the aim of *E. coli* genomics is to reveal a regulatory network with thousands of elements, but the genome is far too complex for this purpose.

To discover genetic essentiality, reducing the genetic elements has been challenged to a large extent. Significant efforts have been made to construct a minimal genome containing the essential genes for growing under the defined conditions to provide the genetically simplest model. There are two main approaches to constructing a minimal genome: bottom-up and top-down. The bottom-up approach is represented by landmark works that chemically synthesize a genome containing only essential genes and transfer it to the cytoplasm [14,15,16]. The top-down approach involves genetic deletions by removing redundant DNA sequences from the wild-type (full-length) *E. coli* genomes, known as genome reduction (Figure 1). Such genetic deletion approaches were employed to a great extent to identify the minimal genetic requirement for a bacterial cell.

So far, the reduced genomes were mainly constructed with the *E. coli* strains of MG1655 and W3110, two representative full-length genomes of approximately 4.6 Mb. Multiple deletions of genomic sequences were commonly performed with the traditional genetic methodologies, e.g., λ Red recombinase and P1 transduction, which were conducted many years ago [17,18]. After the CRISPR/Cas9 system was developed and used widely [19,20], the random deletion method combining CRISPR/Cas9 and transposon also became available to reduce the genome [21]. Comparative genomics and computational approaches are also available for investigating the genome range of reducible [22]. Various genome-reduced *E. coli* strains have been successfully constructed in laboratories [23,24,25,26], and those of extensive deletions (i.e., more than 10% of the parent wild-type genomes are absent) are summarized (Table 1).

The minimal genomes were considered beneficial for material production and genetic recombination. For instance, the successful downstream applications of the reduced genomes summarized in Table 1 were reported. The genome-reduced *E. coli* strain MGF-01 showed increased L-threonine production compared to the wild-type strain W3110 carrying the full-length genome [28,29]. The reduced genome MDS42 was constructed by removing the IS elements, which made it highly useful for genetic recombination without IS-mediated mutagenesis. The efficiency of DNA transformation of MDS42 was over 180-fold higher than that of its parent strain MG1655 and was equal to or higher than that of commercially available competent cells [24]. The recombinant Isoamylase derived from *Thermobifida fusca* has been successfully produced using this strain [30], and the modified strain MDS-205 has improved L-threonine production [31]. Genome reduction of *Bacillus* is also a well-known application of the bacterial minimal genome. As a representative example, *Bacillus subtilis* PG10 has about 36% of its genome deleted, which is industrially applicable with its enhanced protein secretion capacity [32,33]. Additionally, the genome-reduced *Bacillus amyloliquefaciens* strain GR167, lacking approximately 4.18% of the genomic sequence of the parent strain LL3, was constructed and successfully produced 311.35 mg/L of surfactin, a biosurfactant [34].

Deleting the redundant DNA sequences, which are supposed to be unnecessary for living cells, causes differentiated changes in growth fitness. In particular, the reduced genomes of large deletions showed a decrease in growth rate compared to their parental strains under nutritionally poor media (Table 1). The high-throughput growth assay of the reduced genomes found the growth rates were correlated to the deleted length of genomic sequences [35]. That is, the genome’s fitness is somehow connected with its size. In addition, genome reduction is likely to cause an increase in spontaneous mutation rate in a deletion size-correlated manner [36,37]. These studies demonstrated that genome size, growth rate, and mutation rate were quantitatively associated, suggesting that the abundance of genetic information, i.e., genome, is highly coordinated with the fitness and evolvability of the living cells (Figure 2). Even genomic sequences that are nonessential for growing in normal conditions, e.g., rich media, play a role in accelerating cell growth. Genome reduction without growth decrease is crucial for addressing the question of what the minimal genome is. Besides the synthetic approach, i.e., conventional genetic construction, the evolutionary and environmental approaches should be considered to acquire extensively reduced genomes.

### 2.2. Evolutionary Approaches for Reduced Genome

In wild nature, free-living bacteria were found to show high growth fitness despite holding small genomes, such as *Pelagibacter ubique*, an abundant living organism in seawater. It can grow quickly, even with a considerably small genome size of only 20% of that of *E. coli* (e.g., 1 Mb compared to 5 Mb) [39,40]. Intriguingly, the *Pelagibacter ubique* genome contains few transposons or pseudogenes, indicating the reduced genome could have high fitness. It was considered that the living cells had acquired their current fitness through natural evolution. Evolution can be a powerful tool in searching for an essential set of genes to gain improved fitness in the habit (i.e., living environment). Experimental evolution mimics nature evolution under well-controlled laboratory conditions [41,42]. It is generally conducted through the serial transfer of the bacterial populations to select derivatives with improved fitness. The serial transfer is repeated inoculation and dilution of a portion of the bacterial population grown to the early exponential growth phase into a fresh medium with the same composition (Figure 3A). The end-point (evolved) population finally achieves higher fitness than the initial population (ancestor) (Figure 3B), so it is also called adaptive laboratory evolution (ALE) [43]. The improved fitness of evolved bacterial populations was usually accompanied by the accumulation of genome mutations during evolution. Interestingly, when the experimental evolution of an identical ancestor was performed with multiple lineages under the same condition in parallel, the genome mutations found in the final evolved populations often differed among the lineages [44,45]. The genomic location of the mutations and the timing of the mutations fixing on the genome varied among the parallelly evolved lineages [46,47]. It indicates that the experimental evolution provides multiple trajectories for the ancestral genome to acquire somehow finetuned genomic sequences promising growth fitness. So far, intensive studies have demonstrated that experimental evolution is a powerful approach to selecting a bacterial population (genotype) with an increased growth rate (relative fitness) [42,48].

Experimental evolution of reduced genomes has been reported with different *E. coli* strains. The reduced genomes somehow show differentiated evolvability compared to the wild-type genomes. As the deleted size of the genome correlated with the increased mutation rate (i.e., improved evolvability) [37], the removal of IS elements or disabling other mutation-generating pathways of the host could result in reduced evolvability [50,51]. Although, the adaptive evolution to the new environmental conditions could be successfully achieved independent of their full-length parent genomes, e.g., MG1655 and W3110 [45,49,52]. Four reduced genomes lacking 200~1000 kb DNA sequences were evolved for ~1000 generations, and the growth rates of these reduced genomes were raised to an equivalent level to that of the wild-type (full-length) genome [49]. These results show that the decreased growth rate caused by the deletion of a large gene set can be complemented by introducing a few small mutations, e.g., single-nucleotide substitutions, without inserting additional genes.

In addition, the changes in mutation rates were coordinated with the increase in growth rates [37,38], revealing that the experimental evolution compensated for the genome reduction-mediated changes in growth and mutate rates. Note that such coordination might be stringently related to transcriptome reorganization. Although the gene expression patterns were significantly disturbed by genome reduction and experimental evolution, the chromosomal periodicity and negative epistasis (i.e., canceling effect) of the transcriptomes were observed [53]. In a word, experimental evolution rescued the growth and mutation rates disturbed by genome reduction (Figure 4). Intriguingly, the changes in growth fitness caused by either genome reduction or experimental evolution were dependent on the deletion size [37,49], indicating the abundance of genetic information but not the specific gene function participate in the coordinated relationships. Although the underlying mechanisms are unclear, maintaining the homeostatic chromosomal architecture must be crucial for optimizing the growth rate of cells with reduced genomes.

## 3. Environmental Requirement for Minimal Genome

### 3.1. Culture Medium

Discovering the minimal genome by either bottom-up or top-down approaches has put great effort into the genetic issues. The minimal genome must be a growing system, as it is a living cell. Cell growth depends on both genetic and environmental conditions, and the environment can alter the genetic state [54]. The environmental constraint is required to be addressed to discover the maximal removable size of the *E. coli* genome. The chemical condition for cell growth, i.e., the culture medium or growth medium, is lately analysed in a high-throughput manner [55,56], and some reports utilize a database [57]. The culture media comprise various chemical components designed to support cell growth. They are categorized as the minimal media containing only the essential chemicals for cell growth [58,59,60] and the rich media comprising natural ingredients for fast growth or maximal biomass [61,62,63]. The culture medium is tightly related to the genetic requirement via metabolism (Figure 5). Genome reduction may cause the cells to fail to biosynthesize the chemicals essential for growth, e.g., metabolites, gene products, etc., known as auxotrophic [64]. Supplementation of these chemicals in the medium allows the genome-reduced cells to be alive.

Moreover, chemical condition determines not only the fate of cells (alive or dead) but also their fitness (growth fast or slow). Although bacterial growth is usually faster in rich media than in minimal media, the best medium composition is genetically dependent [65]. Genome reduction-dependent decrease in growth fitness was significant in minimal media but tiny in rich media [35]. In addition, the responsivity of genome-reduced *E. coli* strains to environmental changes likely differed from that of the wild-type strains [53]. These findings indicate that the reduced growth fitness caused by genome reduction can be compensated by medium optimization, e.g., changes in medium components or their abundance (Figure 5).

### 3.2. Medium Optimization

Medium optimization is commonly performed to adjust the medium composition for improved cell culture. Since carbon, nitrogen, and phosphate sources are the major cellular nutrients, the relationship between their concentrations and cell growth has been characterized [66,67,68,69]. Because these starvations can cause an arrest of cell growth [70,71,72,73,74], these nutrients tend to be adjusted primarily in the improvement of cell culture and in industrial material production. Metal ions affect cell growth related to osmotic pressure, electric charge, and other factors [69,75,76,77]. To adjust the concentration of medium components to be optimal for cell growth, the method of One-Factor-At-a-Time (OFAT) [78] has been classically used. This method can change the concentration of only one medium component at a time, and its advantage is the convenience and ease without requiring statistics. To optimize more than two medium components, the combination of the statistical method Response Surface Method (RSE) and Design of Experiments (DoE) [79,80,81] is commonly used in laboratories. Plakett Burman Design (PBD) [82,83], Taguchi method [84,85], and Central Composite Design (CCD) [86,87] are available for DoE, and experiments can be conducted based on these designs, and optimal points can be searched by RSE based on experimental results. Despite the convenience and simple manipulation, only a limited number of components can be subjected to medium optimization with these methods. The adjusted level of the first component might be resettled while adjusting the second one [88]. Finetune of all medium components simultaneously is highly essential for achieving an optimized medium supporting the growth of a reduced or minimal genome.

## 4. Machine Learning-Based Minimal Genome Methods

### 4.1. Machine Learning-Assisted Medium Optimization

Machine learning is a computational approach to apprehending representative patterns from a dataset of unknown structures. It has been used in various fields of biology, e.g., genetic prediction, protein engineering, and metabolic engineering [89,90,91,92,93,94,95], as it benefits biological studies from a system point of view. The advantage of machine learning-assisted medium optimization is the ability to predict the appropriate concentrations of medium components independent of personal professional experience or prior literature knowledge, according to a given dataset acquired experimentally. Often, high-throughput methodologies are employed to obtain the cell culture dataset, which links hundreds of medium combinations to the corresponding growth fitness. Applying machine learning algorithms to the dataset captures the data pattern. It constructs the machine learning model, which can predict growth fitness according to the artificial values of medium combinations that are not experimentally tested.

Machine learning-assisted medium optimization often constitutes two procedures. Black-box or white-box machine learning algorithms can be used, which are selected according to whether the output is interpretable. Here, white-box machine learning using the decision tree algorithm is introduced to give an interpreted output as an example (Figure 6). The first step is the learning process, i.e., data mining, which constructs the machine learning model and searches for the medium components deciding the growth fitness. The decision tree algorithms have been successfully used to find the decision-making components in the complicated culture medium for bacterial growth and substrate production [96,97,98]. The second step is to predict the medium composition leading to improved growth fitness according to the model constructed in the first step. It predicts the growth fitness upon novel medium compositions made artificially and selects those in which the predicted growth fitness is significantly improved. Machine learning prediction for better activity has successfully improved the cellular activity of mammalian cells [99], the translation activity of bacteria or cell-free systems [100,101], and the activity of bacterial secondary metabolism [102]. The successful applications strongly suggest that machine learning-mediated methodologies can benefit genome reduction or minimisation.

### 4.2. Active Learning

Machine learning-based methods have the advantage of requiring no prior knowledge or hypotheses; nevertheless, they require a large amount of data to obtain the potential input data patterns and correct labels. The number of factors in the input data (e.g., the number of medium components) increases, then the data required for training increases explosively. It is a severe disadvantage for biological experiments, which require a large number of factors to target and significant time and effort to obtain data. The active learning methodology can be applied to save experimental time and labour. Active learning is a method that not only allows the learned model to make the final prediction but also allows the model to choose what input data (i.e., experiments) are needed to train the model efficiently [103,104]. It combines experimental validation and machine learning prediction in a repeated manner. In the case of medium optimization, the original dataset is first obtained through experiments to build the first predictive model, then make predictions for several untested medium compositions. These predicted medium compositions are experimentally tested, and the results are added to the initial dataset for the following round of model construction and prediction (Figure 7A). Repeated rounds of active learning improve the prediction accuracy of machine learning and the growth fitness in the finetuned medium. The availability and efficiency of active learning for medium optimization have been demonstrated in the case of the mammalian cell [99]. Active learning has also been utilized in drug discovery [105], structural biology [106], and translational activity in cell-free systems [101]. The success in the research areas on gene expression prediction [107] and functional genomics [108] strongly suggests that active learning could be an available tool in genome reduction and minimal genome exploration.

## 5. Perspectives

Since previous studies showed that the medium composition partially compensated for the genetic information [35,49], it is intriguing whether medium optimization could fully offset the genetic deficiency caused by genome reduction. Machine learning-assisted methodologies that take the genetic design and its growth into account simultaneously may need further development. Its application to bacterial cells for pervasive genome reduction must be practical and easily manipulated; for instance, active learning for increased growth fitness of genome-reduced *E. coli* strains facilitates extensive deletions. In addition, combining in silico metabolic prediction with the medium composition might be challenging to search for a minimal genome. The landmark study predicted the metabolism of an intracellular symbiont *Buchnera* with a heavily reduced genome, which was derived from *E. coli* [109]. The essentiality of genes can be spurious, as a gene can easily become essential or nonessential in changing genomic and environmental contexts.

As the essentiality of genetic information is crucially dependent on the medium composition, combined consideration of genome reduction and medium optimization is essential to address the requirements of the minimal genome. Investigating the requirements of the minimal genome in coordination with its growing environment using machine learning techniques may provide us with an optimized “blueprint for a minimal cell” (Figure 7B). The approach comprises two directional processes: finetuning the medium composition for a given genome to promise its growth fitness and searching for the reduced genome maintaining its growth fitness in a given medium. Considering these two processes cooperatively with machine learning must be crucial for designing a minimal cell to avoid significant damage or lethality caused by the genetic deficiency in an alternative suboptimal medium. Despite the vastly accumulated genetic information, its linkage to growth fitness remains insufficient. Despite the countless receipts of media used for decades, their contributions to growth fitness are out of the quantitative evaluation. Future studies to collect the quantitative datasets that connect growth fitness with genetic information (e.g., genome and gene function) and environmental information (e.g., medium compositions) are essential for functional and productive applications of machine learning techniques.

In summary, applying machine learning to find the minimal requirement of a living cell is practical. Linking the genetic or genomic information accumulated in the vastly extensive databases to the quantitatively evaluated growth data will allow us to address the question of the minimal cell without knowing the underlying mechanism that participated in complex living systems. Future studies that combine the genetic and environmental essentiality for the minimal genome or cell are highly required.

## Figures and Tables

**Figure 1 biology-12-01170-f001:**
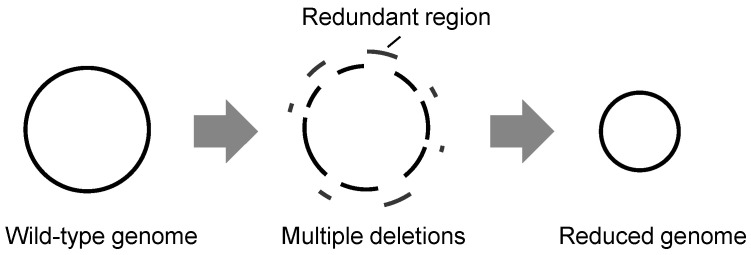
Schematic drawing of genome reduction. A reduced genome is constructed by removing redundant genomic sequences from the parent genome (wild-type genome).

**Figure 2 biology-12-01170-f002:**
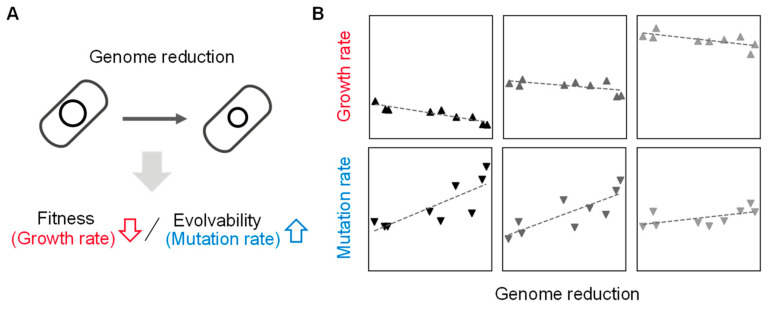
Coordination of genome size to growth and mutation rates. (**A**) Contribution of genome reduction to fitness and evolvability. The changes in growth and mutation rates caused by the genome reduction are indicated with arrows. (**B**) Relative values of genome, growth, and mutation. Gray gradation indicates the variation in growth media. The scatter plots are newly made using previously reported data [37,38]. The panels from left to right represent the nutritional richness of culture media from poor to rich.

**Figure 3 biology-12-01170-f003:**
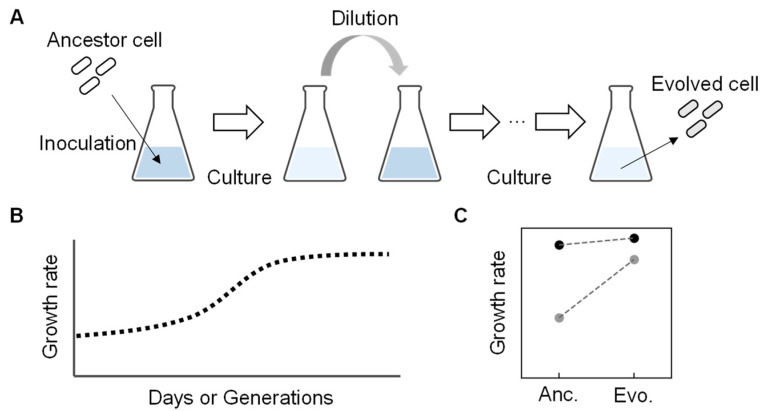
An overview of experimental evolution. (**A**) Experimental evolution. Repeated culture and dilution are performed with the ancestor to acquire the evolved genome with an improved growth rate. (**B**) Temporal changes in growth rate during experimental evolution. (**C**) Growth rate changes between the ancestor and evolved *E. coli* cells. Dark and light circles represent the full-length and reduced genomes, respectively. The previously reported data [49] were used to make the graph.

**Figure 4 biology-12-01170-f004:**
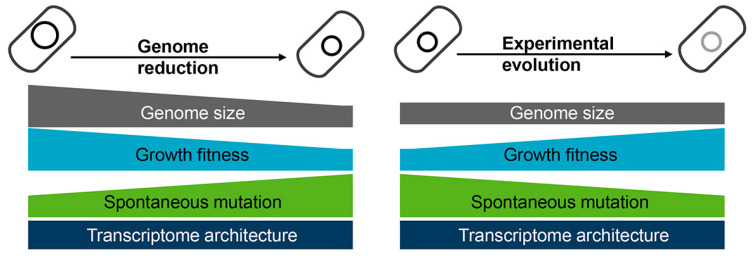
Changes caused by genome reduction and rescued via experimental evolution. Genome reduction causes decreased growth fitness and increased mutation rate (**left panel**), which is restored via experimental evolution (**right panel**). The transcriptome architecture maintains homeostasis regardless of genome reduction or experimental evolution.

**Figure 5 biology-12-01170-f005:**
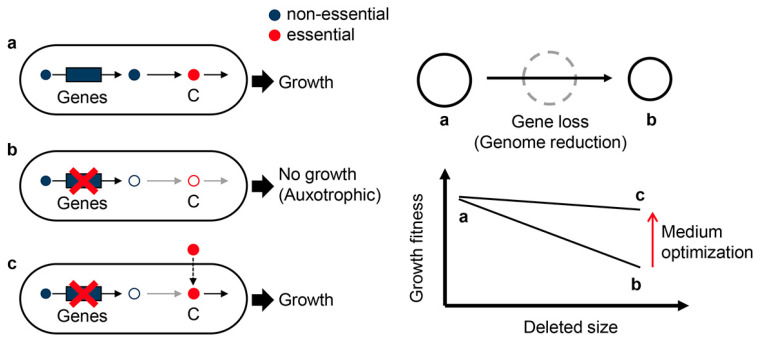
Compensation of medium composition to gene loss. The wild-type strain (full-length genome) grows regularly (a). Gene loss caused by genome reduction leads to nutritional auxotrophs when the genes are essential (b). However, once the component (C) in response to the absent metabolite (C) is supplied in the medium, the growth fitness recovers (c). Decreases in growth fitness associated with the accumulation of genetic deletions can be avoided by changes in environmental conditions, i.e., medium optimization (right panel).

**Figure 6 biology-12-01170-f006:**
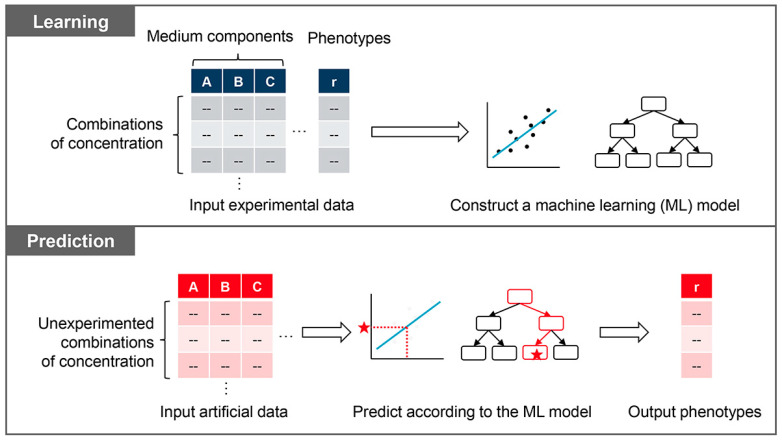
Scheme of machine learning-assisted medium optimization. The learning and prediction processes are illustrated in the upper and bottom boxes, respectively. The learning process constructs a model by training the machine learning algorithm with the experimental dataset that links medium combinations to growth fitness. The prediction process outputs the predicted growth fitness using the constructed model by inputting the novel data generated artificially.

**Figure 7 biology-12-01170-f007:**
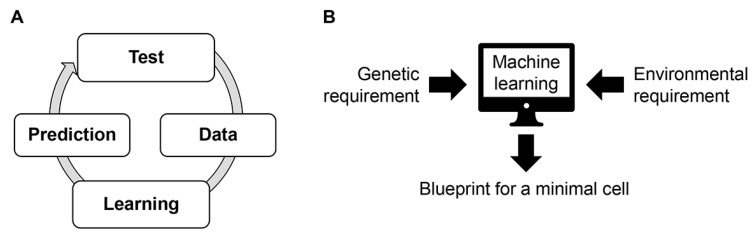
Proposed future challenges. (**A**) Active learning. Repeated experimental tests and machine learning improve prediction accuracy and advanced medium or genome combinations. (**B**) Blueprint for a minimal cell. Combination of genetic and environmental information benefits.

**Table 1 biology-12-01170-t001:** Representative genome-reduced *E. coli* strains. The wide-type strains used as the parent genomes for multiple deletions, the resultant reduced genomes, and their growth fitness in different growth media are summarized according to the previous report [27].

Parent Genome	Strain Name	Genome Size (Mb)	Reduced Ratio	Growth Medium	Growth Fitness
W3110 (4.66 Mb)	MGF-01 (N28)	3.6	22%	minimal	decreased
				minimal, amino acids	decreased
				rich	decreased
	DGF-298	3.0	36%	rich	increased
MG1655 (4.64 Mb)	MDS42	4.0	14%	minimal	equivalent
				rich	equivalent
				minimal, amino acids	decreased
	MDS69	3.7	20%	rich	decreased
	Δ16	3.3	30%	minimal	decreased
	MS56	3.6	23%	minimal	increased, decreased
				rich	equivalent, decreased

## Data Availability

Not applicable.

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
