# Peer review of "Efforts to Minimise the Bacterial Genome as a Free-Living Growing System"

_biology, 2023, doi:10.3390/biology12091170_

Round 1

Reviewer 1 Report

In this manuscript, Honoka Aida et.al., summerized and discussed the genome reduction topic in bacteria. This mini review provided important information. The manuscript was written in a clear and logical manner, the discussion are thorough. However, as a reviewer, I have to say that there are several major concerns need to be addressed before this manuscript can be considered for acceptance.

1, the '3.2 Machine learning-assisted medium optimaization' section and Figure 7 can expand some content of "machine learning based minimal genome methods" and become an independent section.

2, It would be better if add some downstream application researches based on minimal genome.

3, the Figure 2A is somewhat difficult to understand what it means.

4, The table style can be improved for easier reading.

The manuscript was written in a clear and logical manner.

Author Response

The point-to-point response is attached as a pdf.

Reviewer 2 Report

The mini review by Aida and Ying represents an interesting contribution to the field of the study of cells with reduced genomes using Escherichia coli as a model. It states that a correlation exists between the size of the reduced genome, the reduction of growth rate and an increased mutagenesis. However, there are some issues that need to be addressed before it is accepted for its publication.

The data supporting that mutation rate is inversely related to the size of the reduced genome is not clearly shown. Figure 2B that is supposed to show this inverse correlation is not clear at all. I suggest that instead of showing a graphic of genome size, growth rate and mutation rate, it is separated in two graphics one showing genome reduction vs growth rate, and a second one showing genome size vs mutation rate. It would also be interesting to have an additional figure that presents one or two examples of strains with different deletions, growth rates and mutation rates, and how the fitness was increased by experimental evolution.

Other minor suggestions are:

1.     Line 64: Insert "the aim of" between "words," and "E. coli"

2.     Line 90 (Table 1): Interchange the last two columns (Growth fitness and Growth medium)

3.     Line 119: Change “to improve their fitness” to "to select derivatives with improved fitness"

4.     Line 130: Change “was” to “is”.

5.     Line 131: Change "acquiring" to "selecting"

6.     Line 131: Change "of" to "with"

7.     Line 144: Change "It strongly suggest" to "these results show"

8.     Line 156: Change "It strongly suggest" to "these results show"

9.     Line 157: Change "participating" to "participate"

10.  Line 159: Change "achieving the minimal genome as a living system" to optimizing growth rate of cells with reduced genomes"

11.  Line 270: Delete "the" between “avoid” and “significant.

The manuscript is fairly well written, but it will benefit to the review from a native English speaker.

Author Response

(The authors gave the same response as above.)

Reviewer 3 Report

Review of "Efforts to minimise the bacterial genome as a free-living growing system" for MDPI Biology, on 5 August, 2023

In general, I like this approach, and think it makes sense and is a good method to optimize genome reduction in E. coli, whilst also not sacrificing reproductive rates.  Ideally, for synthetic biology, something that can divide rapidly but is also smaller would be great.  But I wonder whether alternative organisms might have already solved this - such as Peligobacter unique, which has only 20% of the genes in a typically E. coli, but can still divide rapidly and is very abundant (hence the 'ubique' in the name!)

A few small comments on the manuscript, with line numbers, is below...

Page 1, line 36 - the smallest viruses can be only a thousand nt long!  Abbreviation should be Kbp (large "K")

Page 1, line 36 - human cells are diploid, so there's 6.4 Gb (not 3.2)

page 1, line 37 - the largest plant genome (so far!) is Paris japonica, at 597 Gb (see https://cvalues.science.kew.org/)

page 1, line 37 - the largest bacterial genome is 16 Mb (not 1.6! Even E. coli is typically ~5 Mb)  see DOI 10.1099/ijs.0.068270-0

page 2, line 56 - the genome size of the FIRST E. coli genome published was 4.6 Mb, but the SECOND E. coli genome published was about 5.6 Gb (see https://www.ncbi.nlm.nih.gov/datasets/genome/?taxon=562) and most E. coli genomes, on average are around 5 Mb, ranging from 3.8 Mb to more than 6 Mb

page 3, line 85 - CRI_S_PR/Cas9

page 4, line 117 -  I like the idea of using evolutionary approaches for reduced genomes. Just as an aside, from an evolutionary point of view, cousins of E. coli have been reduced by 90% (!) and metabolic models can start with E. coli and randomly throw out genes, and accurately predict the metabolic needs for these reduced genomes (see https://doi.org/10.1038/nature04568)  

I also think that 'natural evolution' has already done this.  For example, as an alternative to E. coli, there are other bacteria that can grow quickly, and are quite abundant, such as Pelagibacter ubique, which has a genome size of about 20% of E. coli (e.g., 1 Mb compared to 5 Mb), and can grow quite quickly, and is very abundant in seawater (https://www.science.org/doi/10.1126/science.1114057; https://doi.org/10.1093/molbev/msr203)

There are only a few typos, in general the English is quite good.

Author Response

(The authors gave the same response as above.)

Round 2

Reviewer 1 Report

Your revision has successfully addressed all the concerns I had regarding the manuscript. As a result, I am pleased to confirm that the manuscript is now suitable for acceptance for publication.